# Super-Enhancers, Phase-Separated Condensates, and 3D Genome Organization in Cancer

**DOI:** 10.3390/cancers14122866

**Published:** 2022-06-10

**Authors:** Seng Chuan Tang, Udhaya Vijayakumar, Ying Zhang, Melissa Jane Fullwood

**Affiliations:** 1Cancer Science Institute of Singapore, National University of Singapore, Singapore 117599, Singapore; sc.tang@nus.edu.sg (S.C.T.); e0384128@u.nus.edu (U.V.); csiyz@nus.edu.sg (Y.Z.); 2School of Biological Sciences, Nanyang Technological University, Singapore 637551, Singapore; 3NUS Centre for Cancer Research, Yong Loo Lin School of Medicine, National University of Singapore, Singapore 117599, Singapore; 4Institute of Molecular and Cell Biology, Agency for Science, Technology and Research (A*STAR), Singapore 138673, Singapore

**Keywords:** super-enhancers, chromatin looping, phase-separated condensates, 3D genome organization, cancer, drugs targeting chromatin interactions

## Abstract

**Simple Summary:**

Gene expression is primarily controlled at the level of transcription, largely due to binding of transcription factors to specific sites on DNA, namely super-enhancers and promoters. Super-enhancers and enhancers can regulate distal target genes via chromatin looping mechanisms. These long-range enhancer–promoter interactions are often mediated by phase-separated condensates. In this review, we summarize recent insights into super-enhancers and phase-separated condensates and their roles in the transcriptional activation of genes involved in development and diseases such as cancer. In addition, we survey a broad array of drugs that target super-enhancers, three-dimensional genome organization, and phase-separated condensates that could be potentially used for the treatment of cancer.

**Abstract:**

3D chromatin organization plays an important role in transcription regulation and gene expression. The 3D genome is highly maintained by several architectural proteins, such as CTCF, Yin Yang 1, and cohesin complex. This structural organization brings regulatory DNA elements in close proximity to their target promoters. In this review, we discuss the 3D chromatin organization of super-enhancers and their relationship to phase-separated condensates. Super-enhancers are large clusters of DNA elements. They can physically contact with their target promoters by chromatin looping during transcription. Multiple transcription factors can bind to enhancer and promoter sequences and recruit a complex array of transcriptional co-activators and RNA polymerase II to effect transcriptional activation. Phase-separated condensates of transcription factors and transcriptional co-activators have been implicated in assembling the transcription machinery at particular enhancers. Cancer cells can hijack super-enhancers to drive oncogenic transcription to promote cell survival and proliferation. These dysregulated transcriptional programs can cause cancer cells to become highly dependent on transcriptional regulators, such as Mediator and BRD4. Moreover, the expression of oncogenes that are driven by super-enhancers is sensitive to transcriptional perturbation and often occurs in phase-separated condensates, supporting therapeutic rationales of targeting SE components, 3D genome organization, or dysregulated condensates in cancer.

## 1. Introduction

The regulation of gene expression in eukaryotic cells is a coordinated biological process that relies on the efficient assembly of transcription factors (TFs), transcriptional co-activators such as Mediator complex subunit (MED1) and BRD4, and RNA polymerase II (RNA Pol II) at specific genomic sites (Figure 1) [1]. Here, we review recent advances in our understanding of the roles of super-enhancers (SEs) and phase-separated condensates in the transcriptional activation of genes involved in cancer. In particular, we survey the broad array of drugs that target SEs, chromatin interactions, and the phase-separated condensates that have been identified so far. We highlight key areas for future investigation and speculate on how these drugs can be exploited for the treatment of cancer.

## 2. Constituents and Identification of Super-Enhancers

SEs are clusters of neighboring enhancers spanning over 10 kb with high-fold enhancer activity that drive cell-type specific gene expression [2]. 3D genome organization enables SEs to interact with specific gene promoters and orchestrates their activity as evidenced by the high frequency of chromatin interactions at the genomic loci containing SEs [3]. SEs contain many TF binding sites, and are heavily loaded with enhancer-associated chromatin features, such as master TFs (e.g., Oct4, Sox2, Nanog, and Klf4 in embryonic stem cells [4]), RNA Pol II, MED1, and chromatin modifiers (p300 and BRD4). The recruited factors alter the chromatin structure, leading to interactions with promoters and RNA Pol II, a process mediated by enhancer–promoter looping (Figure 1). Phase separation may facilitate the assembly and function of SEs [5].

SEs are defined by ranking the enriched ChIP-seq signals of master TFs, co-activators BRD4, MED1, p300, or enhancer-specific histone marks, such as H3K27ac and H3K4me1 [6,7,8]. To facilitate SE identification in silico, approaches such as ROSE were developed, in which the adjacent enhancers are stitched based on H3K27ac ChIP-seq peaks, and SEs are separated from typical enhancers by the inflection point present in ranked H3K27ac ChIP-seq signals plot. Top-ranking enhancers above the transition point are designated as SEs [4]. SEs have a higher order of magnitude of transcriptional factor density, size, and ability to activate transcription compared to the typical enhancer [4]. For example, although both SEs and typical enhancers are occupied by master TFs such as Oct4 and Nanog, SEs are more densely occupied by TFs, which are crucial for mESC identity such as Klf4. The histone modification levels at SEs exceeded the typical enhancers by, at least, an order of magnitude [4].

These SE profiles can be used to identify master genes that are involved in cell identity and disease condition [2,4]. Additionally, SE profiling helps us to identify potential drug targets in various cancer types. For example, by profiling SEs in t(4;14)-translocated multiple myeloma, SE-associated gene HJURP was identified as a potential target as its silencing impaired cell growth and induced apoptosis [9]. Additionally, through mapping SEs in Acute Myeloid Leukemia (AML), a subtype of AML cells with SE-driven RARα was identified to be sensitive to RARα agonist SY-1425 [10].

### 2.1. Super-Enhancers and Chromatin Interactions

The human genome is organized into higher order structures, and such structures are important for transcriptional regulation [11]. Individual chromosomes occupy distinct regions of the nucleus, known as chromosome territories, that are themselves spatially segregated in A and B compartments. The A compartment is associated with actively transcribed genes, whereas the B compartment is associated with epigenetically silent genes and gene-poor DNA. Genome-wide Hi-C analysis showed that loci located on the same chromosome interact more frequently than any two loci located on different chromosomes [12]. At the sub-megabase scale, chromatin is compartmentalized into smaller structures known as topologically associating domains (TADs). TADs are self-interacting, loop-like domains that contain interacting *cis*-regulatory elements and target genes [13]. The chromatin fiber is organized into a collection of DNA loops which establish chromatin interactions with distant regions and regulate the activity of genes. This is explained by the loop extrusion model in which frequent transient loops are organised by structural maintenance of chromosomes (SMC) complexes that reel in chromatin, forming growing loops that stop at CCCTC-binding factor (CTCF) boundaries [14,15]. TAD borders are demarcated by convergently oriented CTCF binding sites that obstruct loop extrusion and cohesin translocation. CTCF proteins act as loop anchors and insulate TADs from neighboring regions. Insulated neighborhoods are chromosomal loops, which bound by CTCF homodimers, occupy by the cohesin complex, and contain at least one gene [16,17]. Most of the enhancer–promoter interactions are contained within insulated neighborhoods [18].

Several SE-associated factors, such as the CTCF and cohesin complex mediate chromatin interactions within the SEs [18]. Integrated Hi-C and ChIP-seq data analysis identified enriched CTCF binding, and a higher frequency of chromatin interactions present at hub enhancers within the hierarchical SEs [19]. Thus, CTCFs regulate cell type-specific and cancer-specific SEs [20].

The transcriptional activity of SEs is restricted within insulated neighbourhoods enclosed by CTCFs and cohesin complex such that SEs are specifically tethered to their target genes. Cohesin loss leads to the development of myeloid neoplasms [21]. Higher occupancy of cohesin and CTCF molecules that mediate long-range chromatin interactions and chromatin looping is noted in SE constituents, suggesting the loops connecting SEs and promoters are strictly controlled [22]. In T-lymphoblastic leukemia, SEs targeting the *IL7R* locus are restricted within the same CTCF-organized neighborhood [23]. SEs insulated by strong TAD boundaries are frequently co-duplicated in cancer patients [24].

The disruption of the insulated chromatin neighborhood by deletion of the CTCF binding site at one of the borders causes dysregulation of intradomain genes and activation of genes outside the neighborhood [16]. Functional CTCF occupancy at the borders of the SE domain was validated in the in vivo mouse model [25]. In mammary tissue, mammary-specific *Wap* SE (comprised of three constituent enhancers) activated neighboring non-mammary gene *Ramp3* separated by three CTCF sites. Although CTCF does not completely block SE activity, deletion of CTCF in mice demonstrated the capacity to muffle gene activation. CRISPR/Cas9-mediated deletion of three CTCF sites did not alter *Wap* expression, but increased *Ramp3* expression (seven-fold) in mammary tissue from parous mice by establishing enhanced chromatin interactions between S3 of *Wap* SE and the first intron of *Ramp3* [25]. This indicates that CTCF sites are porous borders instead of tight blocks and they muffle SE-mediated activation of secondary target genes present outside of the insulated neighborhood. Thus, proto-oncogenes can be activated in cancer cells upon loss of the insulated boundary through enhancers present outside the neighborhood [26].

In cancer models, elevated *MYC* oncogene levels are associated with aggressive tumors. One of the ways that this dysregulation is achieved is through the acquisition of large tumor-specific SEs present within 2.8 Mb *MYC* TAD. In tumor cells, SEs at the *MYC* locus are looped to a common CTCF site within the *MYC* promoter (Figure 2A) [27]. CRISPR/Cas9-mediated perturbation of a *MYC* promoter-proximal CTCF binding site in tumor cells leads to reduced chromatin interactions between the *MYC* promoter and distal SEs present downstream of *MYC*, indicating that the CTCF docking site is necessary in mediating enhancer–promoter looping [27]. DNA methylation of these *MYC* enhancer docking site with dCas9-DNMT3A-3L protein and specific gRNA reduced *MYC* expression in K562 and HCT-116 cancer cell lines, possibly due to abrogation of CTCF binding upon methylation [27].

Recently, 3D genome organization in T-cell Acute Lymphoblastic Leukemia (T-ALL) was characterized, in which TAD fusion was observed in the *MYC* locus in T-ALL, subjecting its promoter to chromatin interactions with SE [28]. TAD fusion in the *MYC* locus is associated with increased inter-TAD interaction and the absence of CTCF binding. This fusion brings *MYC* promoter and SE into proximity establishing chromatin interactions that are separated by insulation in normal T cells. Thus CTCF-mediated insulation of TAD determines the accessibility of chromatin looping of the *MYC* promoter with SE. Also, an increase in CTCF binding downstream of SEs was noted, and this could act as super-anchors that mediate SEs and gene interaction [29,30].

Various dCas9 systems, such as dCas9-KRAB [31], dCas9-DNMT3A [31], dCas9-DNMT3A-3L [31], dC9Sun-D3A [32], and dCas9-MQ1 [33], can be used to potentially target methylation of enhancer docking sites and alter CTCF binding to these docking sites. With improvements in the delivery of CRISPR/Cas9 and CRISPR/dCas9 vectors [34,35], targeting oncogenic enhancer docking sites and super-anchors using these vectors could become potential future cancer therapies.

### 2.2. Mechanisms Related to the Acquisition of Super-Enhancers in Cancer

Cancer cells can acquire oncogenic SEs either through chromosomal rearrangements, DNA mutations and indels, 3D chromatin structural changes, or viral oncogenes [36,37,38]. In particular, the disruption of TAD boundaries and dysregulated chromatin interactions can activate oncogene expression. For example, mutations or insertions create a novel binding site for master TFs that recruit other factors and form a strong SE which then activates adjacent oncogenes. Deletion of the CTCF binding site leads to the activation of a silent oncogene by juxtaposed SE. The binding of activation-induced cytidine deaminase triggers genome instability and gene translocation which brings oncogenes near SEs [39]. SEs are exceptionally sensitive to perturbations by transcriptional drugs [40]. A small change in the concentration of components associated with SE activity, such as transcriptional co-activators, causes drastic changes in SE-associated gene transcription [41]. Thus, disruption of the SE-associated gene transcription by targeting these components seems a promising approach for anti-cancer therapy. We discuss methods for targeting these components in Section 2.3.

### 2.3. Targeting Transcriptional Co-Activators and Chromatin Remodelers

Co-activators such as BRDs (BRD2-4, and BRDT) and cyclin-dependent kinases (CDK7, and CDK9) may be targeted to disrupt SEs. Several bromodomain and extra-terminal domain (BET) inhibitors, and CDK inhibitors have been reported to target SEs, as shown in Table 1. For example, treatment of MM1.S myeloma cells with JQ1 (BRD4 inhibitor) leads to preferential loss of BRD4 at SEs and selective inhibition of SE-driven *MYC* transcription [41]. Similar effects were seen in other cancer types such as colorectal cancer [42], ovarian cancer [43], Merkel cell carcinoma [44], B-cell lymphoma [45], and alveolar rhabdomyosarcoma [46].

CDK7 and CDK9 are important in the initiation and elongation of transcription mediated by the phosphorylation of RNA Pol II. CDK7 inhibitor (THZ1) alters the H3K27ac mark globally. In Chronic Myelogenous Leukemia, THZ1 disrupted the transcription of SE-associated gene XBP1 and eradicated leukaemia stem cells [47]. Several cancer subtypes that are sensitive to CDK7 inhibitor, such as oesophageal squamous cell carcinoma [48], triple-negative breast cancer (TNBC) [49], MYCN-dependent neuroblastoma [50], and non-small cell lung cancer [51]. For more details, we refer readers to several excellent reviews that have summarised cancer-specific SEs and potential SE inhibitors.

The ATP-dependent chromatin remodelers consist of the SWI/SNF, ISWI, INO80, and CHD families. SWI/SNF complex is a major regulator of distal lineage-specific enhancer activity [52]. Deletion of this complex in mouse embryonic fibroblasts results in H3K27ac loss and deactivation of the enhancer [52]. SWI/SNF ATPase degradation with AU-15330 (PROTAC degrader of SMARCA2 and SMARCA4) led to disruption of 3D loop interactions of SE with the promoter of *AR*, *FOXA1*, and *MYC* oncogenes, and decreased oncogenic expression in prostate cancer cells [53].

The INO80 complex occupies SEs and drives oncogenic transcription by regulating Mediator recruitment and nucleosome occupancy [54]. Silencing of INO80 results in downregulation of the SE-associated genes and inhibition of melanoma cell growth [54]. The NuRD complex subunit CHD4 localizes to SEs and regulates SEs accessibility to which PAX3-FOXO1 fusion protein binds and activates SE-driven gene transcription in fusion-positive rhabdomyosarcoma [55].

Anticancer drug Lysine-specific demethylase 1 (LSD1) inhibitor (NCD38) activates GFI1-SE and induces lineage switch from erythroid to myeloid by activating differentiation in leukemic cells [56,57]. NCD38 evicts the histone repressive modifiers such as LSD1, CoREST, HDAC1, and HDAC2 from GFI1-SE [57]. Mediator-associated kinases such as CDK8 act as negative regulators of SE-mediated transcription. Mediator kinase inhibitor cortistatin A (CA) inhibits CDK8 and activates SE associate transcription of tumor suppressors and lineage controllers in AML [58]. As both I-BET151 (BET inhibitor) and CA have an opposing effect on SE-associated gene transcription, the authors suggest that cancer cells may depend on the dosage of SE-associated gene expression. The co-treatment did not neutralize the opposing effects but rather inhibited cell growth [58].

BET inhibitors such as FT-1101, RO6870810 (TEN-010), I-BET762, BMS-986158, OTX-015 (MK-8628), ABBV-075, AZD5153, BI 894999, ODM-207, ZEN-3694, PLX51107, NUV-868, TQB3617, and CPI-0610 are under clinical trials for haematological and solid tumors (http://clinicaltrials.gov/). Whereas CDK7 inhibitors such as SY-5609, XL102 and CT7001 are under clinical trials for advanced solid tumors (http://clinicaltrials.gov/). These inhibitors are in clinical trials either being tested alone or in combination with other drugs.

Transcription factor IIH (TFIIH) is a 10-subunit complex (core units XPB, XPD, p62, p52, p44, p34, and p8; dissociable units MAT1, CCNH, and CDK7) that regulates RNA Pol II transcription. Triptolide inhibits XPB subunit of the TFIIH complex and disrupts SE interactions and down-regulated SE-associated genes (*MYC*, *BRD4*, RNA Pol II, and *COL1A2*) in pancreatic cancer [59]. Minnelide (pro-drug of triptolide) is under phase II clinical trial for refractory pancreatic cancer (NCT03117920) and adenosquamous carcinoma of the pancreas (NCT04896073). Oral therapeutic drug GZ17-6.02 (602) comprises a mixture of curcumin, isovanillin, and harmine, that is known to affect the histone acetylation at SE-related genes in pancreatic ductal adenocarcinoma is under Phase 1 clinical trial for advanced solid tumors and Lymphoma (NCT03775525) [60].

### 2.4. Drug Resistance to Super-Enhancer Drugs

Although SE drugs seem to be promising therapeutics, resistance to BRD4 inhibitor JQ1 has been reported in breast cancer and AML. SEs are gained in the established JQ1-resistant TNBC cell lines and are associated with enriched BRD4 recruitment to the chromatin in bromodomain independent manner. Increased expression of SE-associated genes such as BCL-xL makes them resistant to apoptosis compared to the parental cell line. The resistant cells are still addicted to BRD4 and are unaffected by JQ1, as JQ1 could not displace BRD4 from the chromatin due to an increase in stable pBRD4 levels binding with MED1 in resistant cells. This hyperphosphorylation could probably by the decrease in PP2A activity. Combined treatment of JQ1 with either BCL-xL inhibitor (ABT737), CK2 inhibitor (CX-4945) or PP2A activator (perphenazine) could overcome this resistance [73].

Long-term treatment of JQ1 resulted in the activation of drug-resistant genes in breast cancer cells. BRD4 associates with the repressive complex LSD1/NuRD1 and occupies H3K4me1 defined SEs. BRD4/LSD1/NuRD complex then represses the activation of drug-resistant genes such as *WNT4*, *LRP5*, *BRAF*, *GNA13*, and *PDPK1* in breast cancer cells [74]. During long-term treatment with JQ1, the overexpressed PELI1 E3 ligase degrades LSD1, thus decommissioning the BRD4/LSD1/NuRD1 complex. This activates *GNA13* and *PDPK1* expression leading to drug resistance in breast cancer [74]. Combined treatment of BRD4 inhibitor and PELI1 inhibitor (BBT-401) may be effective in treating breast cancer.

By contrast, drug-resistant AML cells show yet another mechanism for acquiring drug resistance, in which *MYC* is activated in the absence of BRD4, possibly by activation of the Wnt pathway [75,76]. BET resistance in AML arises from the leukaemia stem cell population with upregulated Wnt signalling [75]. Despite the loss of Brd4, sustained oncogenic *Myc* expression equivalent to the control cells was observed, as β-catenin occupies the sites where Brd4 is decreased. Inhibition of Wnt signaling resensitizes the cells to BET inhibitors [75]. Similarly, another study reported that PRC2 complex suppression promotes BET inhibitor resistance in AML by remodeling the regulatory pathways and restoring the transcription of oncogenic *Myc* [76]. In response to BET inhibition, focal enhancer formed in established BET resistant cells drives *Myc* expression by recruiting activated Wnt machinery to compensate Brd4 loss. Overall, Wnt signaling acts as a driver and biomarker in acquired BET-resistant leukemia.

In the future, strategies will need to be devised to reduce or delay the development of drug resistance, for example, through combinations of cancer therapies where multiple epigenetic drugs are used to target potential avenues of drug resistance.

## 3. Drugs Targeting 3D Genome Organization

3D genome organization represents another layer in the epigenetic regulation of gene expression. In particular, the disruption of TAD boundaries and dysregulated chromatin interactions can activate oncogene expression, thus causing various diseases, including cancer [77]. One example would be TAD boundary disruption in IDH mutant glioma, where the TAD boundary was disrupted by the elevated methylation at the CTCF site, leading to activation of oncogene *PDGFRA* by contacting the gene with its enhancer outside of the TAD (Figure 2B) [78]. Aberrant chromatin interactions can also drive oncogene expression. For example, cancer-specific *TERT* promoter mutations can recruit transcription factor GABPA to mediate long-range chromatin interactions, thus activating *TERT* expression in melanoma [79]. Therefore, targeting and manipulation of 3D genome organization may become new cancer therapies.

SE drugs have been investigated for their effects on chromatin interactions. For example, the CDK7 inhibitor THZ1 treatment led to the loss of more than half of the chromatin interactions in HK1 cells [80]. THZ1 treatment reduced the contact frequency between *MYC* promoter and its SE in CUTLL1 cells [28], with the underlying possibility that SE drugs may function as genome organization modulating drugs, as well.

It is well established that the 3D genome is highly maintained by several architectural proteins: CTCF, Yin Yang 1 (YY1), and cohesin complex [81,82]. It is speculated that targeting these genome organizers or their interacting partners might be one strategy to disrupt 3D genome organization. The cohesin complex is formed of SMC1, SMC3, RAD21, and SCC3 (SA1 or SA2). Factors that govern cohesin loading onto chromatin (NIPBL-MAU2) and its release from chromatin (WAPL) also affect the function of cohesin complex [83]. Degradation of RAD21, a core component of the cohesin complex, by an auxin-inducible degron system eliminated TADs and loops [84]. Deletion of the cohesin-loading factor NIPBL in mouse liver also led to disappearance of TADs [85].

These pieces of data highlight the importance of cohesin complex in genome folding, and thus, inhibitors related to cohesin complex, loading proteins, and release proteins may be potential drugs for affecting 3D genome organization (for potential cohesin complex inhibitors, see review: Antony et al., 2021) [86]. Some drugs, including glycyrrhizic acid and HDAC8-specific inhibitor PCI-34051, can indirectly affect the function of SMC3 and RAD21 (Table 2) [87,88]. However, no specific drugs directly targeting cohesin complex are developed. In particular, there is no known enzymatic activity that could be inhibited for SA1 [89]. Complete loss of cohesin complex is deadly [86], thus revealing the detailed mechanism by which cohesin affects 3D genome organization is essential to finding new indirect targets to reduce cohesin binding.

YY1 is another well-known chromatin regulator that controls enhancer–promoter loops. Depletion of YY1 can disrupt such interactions [90]. Attempts have been made with YY1 inhibitors including siRNA YY1, nitric oxide donors, proteasome inhibitors, and inhibitors of activated survival pathways such as inhibitors of nuclear factor-kappa beta (see review: Benjamin Bonavida, 2017) [91]. However, there is still a long way to go in terms of developing specific and targeted inhibitors of YY1. Homozygosity for the mutant YY1 allele results in embryonic lethality in the mouse model [92]. YY1 is also a multifunctional transcription factor, which not only contributes to the chromatin interactions, but also functions as the transcription activator or repressor [92]. These facts add to the difficulty of targeting YY1 only for the purpose of affecting chromatin interactions. Therefore, the functions of YY1 in different development stages and cellular contexts need to be elucidated, which helps develop the specific YY1 inhibitors.

CTCF is another architectural protein that is absolutely and dose-dependently required for both chromatin looping and TADs formation [93]. No CTCF inhibitors are yet known to reduce CTCF protein levels and deletion of CTCF proteins is embryonically lethal [94]. However, there are indeed CTCF binding inhibitors, such as curaxin CBL0137, which can function as a 3D genome-modulating drug to target dysregulated gene expression in cancer cells [95]. Curaxins are a class of compounds capable of simultaneously activating p53 and suppressing NF-κB without inducing genotoxicity [95]. The lead curaxin CBL0137 showed significant efficacy in preclinical cancer models, including melanoma [96], glioblastoma [97,98], neuroblastoma [99], and pancreatic cancer [100]. Recently, Kantidze et al. reported that CBL0137 compromised TADs and disrupted the enhancer–promoter chromatin interactions, thereby inhibiting the enhancer-controlled transcription [101]. Specifically, CBL0137 treatment led to decreased TAD boundary strength and enhancer–promoter contacts via decreased binding of CTCF to DNA. Moreover, compared to other known DNA intercalators, CBL0137 is capable of preventing CTCF binding without inducing DNA damage [101]. On this basis, curaxin CBL0137 was classified as a potential 3D genome-modulating drug for anticancer therapy. However, it remains to be seen if the reduction in CTCF binding and modulation of 3D genome organization is indeed the mechanism by which curaxin kills cancer cells, or if it is simply a side effect.

Since CTCF-associated partners also mediate many of the DNA binding and chromosomal organization responsibilities of CTCF, they may be good candidates to target. For example, the *MYC*-associated zinc finger protein (MAZ) has been demonstrated to function as an insulator protein by augmenting the CTCF binding to organize the genome. Depletion of MAZ disrupted the short-range chromatin interactions, as well as decreased the insulation score of TADs, which makes it a promising genome organizer [102].

Topoisomerase II beta (TOP2B) is another well-known CTCF partner that can bind to CTCF to facilitate supercoiling at the TAD boundaries [103]. ZNF143 was recently shown to mediate the enhancer–promoter loops in murine hematopoietic stem and progenitor cells, and the binding ability of CTCF to DNA was dependent on ZNF143 [104]. In addition, SNF2H was also identified as a CTCF partner, and lack of SNF2H affected the chromatin loops and TADs, probably through its requirement for CTCF binding [105]. Besides the above-mentioned factors, CTCF also cooperates with TAF3 and BRG1 to mediate long-range chromatin interactions [106,107]. CTCF-interacting partners and possible drugs targeting these partners are listed in Table 2. In the future, it would be interesting to explore the detailed mechanisms of how these factors affect CTCF functions and genome organization, through generating loss-of-function clones or mutants followed by examining the changes of phenotypes and 3D genome organization. In particular, the dissections need to be done in different cell lines or tissues, as they may be cellular-context dependent.

Besides individually validated CTCF partners, there are some CTCF mass spectrometry data in different cell lines [106,108,109,110,111], pointing out the large list of potential CTCF partners that need to be further explored. For example, through CTCF mass spectrometry in MDA-MB-435 cells, the general transcription factor II-I (TFII-I) was found to interact with CTCF, directing CTCF to the promoter proximal regulatory regions of target genes across the genome, particularly at genes involved in metabolism [108]. However, whether TFII-I was linked to 3D genome organization is unknown. Nucleophosmin, the protein encoded by NPM1, was found to interact with CTCF in HeLa cells by mass spectrometry [109]. It plays a role at known CTCF-dependent insulator sites in vivo; however, the link between NPM1 and 3D genome organization is still unclear. NPM1 is mutated in cancers such as AML, accounting for about 30% of the cases, which makes it an attractive target for future cancer therapies [112]. Together, these CTCF mass spectrometry datasets are potentially rich resources in the search for potential inhibitors disrupting CTCF function and 3D genome organization. In the future, detailed investigation needs to be done to explore the exact relationship between these CTCF partners and CTCF function. In particular, the effects of these partners on 3D genome organization need to be confirmed.

Polycomb repressive complexes (PRC1 and PRC2) also contribute to the 3D genome organization. PRC2 consists of four core subunits EZH1/2, EED, SUZ12, and RbAp46/48, together with various other components, including AEBP2, PCLs, and JARIDS [113]. Gain-of-function mutation of EZH2 (EZH2Y646X) altered the topology [114], and EZH2 inhibition is capable of causing loss of long-range chromatin interactions [115,116]. Plenty of EZH2 inhibitors have been developed (see review: Kim and Roberts, 2016) [117], although it is unclear whether these function through the mechanisms of disrupting 3D genome organization. Similar to EZH2, EED knockout showed loss of extremely long-range chromatin interactions in mouse embryonic stem cells (mESCs) [118], making it a potential target for affecting 3D genome organization. A group of EED inhibitors have been summarized in the recent review [119].

PRC1 complex is formed by CBX (CBX2, CBX4, and CBX6-8), PCGF1-6, HPH1-3, and two RING proteins (RING1 and RING2) [120]. Hi-C data of RING1B knockout mESC cells displayed dramatically altered contact frequency, especially at the HoxA cluster region [121]. In line with this data, double knockout of RING1A and RING1B led to significantly reduced interchromosomal associations between Hox network members [122]. Other PRC1 components, such as CBX2 and CBX5 [123,124,125], have also been associated with 3D genome organization. Therefore, drugs targeting PRC1 complexes (Table 2) might be one direction for 3D genome-modulating drugs. Together, inhibitors directly or indirectly targeting PRC complexes are still on their way to being developed. Although they are indeed promising targets as 3D genome-modulating drugs, in particular, their detailed mechanisms need to be explored. It also remains to be determined if 3D genome organization modulation is indeed the mechanism of action of cancer-specific killing, or if it is just a side effect. Additionally, drug resistance mechanisms to 3D genome organization need to be determined in order to combat the inevitable development of drug resistance.

## 4. Phase Separation Model to Explain Features of Super-Enhancers

Several features of SEs in embryonic stem cells and cancer cells include a high density of TFs and co-activators binding, the robustness of driving gene expression, and the ability to simultaneously activate two independent genes [4,41]. In addition, SEs are more sensitive to drugs that block the binding of BRD4 to acetylated chromatin or CDK7 than typical enhancers [41,50], but these data lack quantitative description. In an attempt to explain quantitatively how SE-driven transcriptional activation operates, a model based on the phase separation of multi-molecular assemblies was proposed by Hnisz and colleagues [5]. Increasing evidence from both in vitro and in vivo experiments supports the idea that phase separation can be used to explain the features of SEs, including aspects of their formation, function, and vulnerability [139,140,141,142].

### 4.1. Cooperative Interactions between Transcriptional Machinery and the Genome

Eukaryotic cells contain membraneless organelles, such as nucleolus, Cajal bodies, stress granules, nuclear speckles, and P bodies that are formed through a process of liquid-liquid phase separation (LLPS). These membraneless organelles are collectively called biomolecular condensates [143]. Phase separation of fluids separates molecules into a dense phase and a dilute phase, where components are rapidly moved into and within the dense phase [144]. Transcription regulation at SEs has been shown to be critically mediated by phase-separated condensates [139].

Transcriptional condensate formation at specific genomic loci is a cooperative process, involving a combination of specific structured TF-DNA interactions and weak multivalent interactions between intrinsically disordered regions (IDRs) of TFs, co-activators (e.g., MED1 and BRD4), and RNA Pol II (Figure 1) [139,140,141,142]. Ahn et al. (2021) demonstrate that IDRs harbored within NUP98-HOXA9, an oncogenic transcription factor chimera recurrently found in a subset of leukemias, form phase-separated condensates to enhance the binding of NUP98-HOXA9 TF to its genomic targets and facilitate long-distance enhancer–promoter looping at proto-oncogenes (Figure 2C) [145]. Sequence-specific TF comprises a DNA-binding domain and an activation domain (AD). The IDRs in its AD are thought to interact with other proteins to mediate the formation of condensates. Boija et al. further showed that the pluripotency transcription factor OCT4 forms phase-separated condensates with MED1 through its ADs to activate genes [139].

To link the enhancer DNA sequence and the formation of phase-separated condensates, Shrinivas et al. demonstrated that the DNA sequences encoding TF binding site number, density, and affinity must exceed sharply defined thresholds for transcriptional condensates to form at specific genomic loci that function as enhancers [142]. Sabari et al. demonstrated that the IDRs of MED1 form phase-separated droplets that can compartmentalize and concentrate BRD4 and RNA Pol II from a nuclear extract [141]. Using live cell super-resolution and light sheet imaging, Cho et al. reported that Mediator and RNA Pol II form condensates that are chromatin-associated and occasionally colocalize with the SE-controlled *Esrrb* gene in mouse embryonic stem cells [140].

The transcription factor YY1 is overexpressed in prostate [146], colon [147], liver [148], lung [149], and breast cancers [150], and its over-expression is associated with poor prognosis in patients with osteosarcoma and Acute Lymphoblastic Leukemia [151,152]. The histidine cluster in the AD of YY1 compartmentalizes BRD4, MED1, and enhancer elements in phase-separated condensates to activate FOXM1 expression in breast cancer [153]. Besides AD of TF, estrogen can bind to ligand-binding domains of estrogen receptor α (ERα) to form MED1 condensates [139].

It is still unclear how transcriptional condensate formation may contribute to long-range chromatin interactions or higher-order chromatin organization. Shrinivas et al. proposed a model in which transcriptional condensates at SEs are formed by the universal cooperative mechanism of phase separation (Figure 1). In this model, a series of biochemical steps have to occur, such as TFs binding to DNA, and this binding must exceed a sharply defined threshold before transcriptional condensates form at specific loci that function as enhancers. This model could also be used to explain why at certain high-affinity TF sites with low local density (such as those that are not enhancer sites), transcriptional condensates cannot form [142].

However, this model does not clarify the sequence of events underlying the formation of long-range chromatin interactions or the formation of transcriptional condensate. It has been proposed that the chromatin loop extrusion mechanism, which is regulated by the insulator protein CTCF and cohesin complex, drives genome 3D organization [154,155]. Lee et al. showed that chromatin looping mediated by CTCF provides a local structural hub for the formation of transcriptional condensates composed of RNA Pol II, MED1 and BRD4 [156]. However, the role of cohesin in mediating the formation of transcriptional condensates is unknown. One possible approach is to use auxin-inducible technology to allow rapid depletion of RAD21, a subunit of cohesin complex and then perform dual-color super-resolution imaging of RAD21 and RNA Pol II clusters in cell nucleus would shed light on the role of cohesin in mediating the formation of transcriptional condensates.

### 4.2. Regulation of Transcriptional Condensates

The concentrations and chemical modifications of RNA molecules have been shown to play a regulatory role in the formation and dissolution of condensates. Henninger et al. proposed a non-equilibrium RNA feedback model in which lower concentrations of short RNAs produced during transcriptional initiation enhance condensate formation, whereas increasing concentrations of the longer RNAs produced during elongation result in condensate dissolution [157]. These results suggest that local RNA levels play a role in the dynamics of transcriptional condensates. In addition, Lee et al. showed that RNA modification such as N6-methyladenosine (m6A) methylation, is highly deposited to nascent RNAs including promoter upstream antisense RNAs and enhancer RNAs (eRNAs) produced during the transcription process in breast cancer cells. These m6A-marked ncRNAs recruit the m6A reader YTHDC1 to partition into liquid-like condensates, which facilitate the formation of transcriptional condensates and therefore gene activation [158].

The post-translational modification of proteins involved In the formation of condensate dictates the condensates in which they associate. For example, the C-terminal domain of RNA Pol II is subjected to phosphorylation by CDK7 or CDK9 that alters the partitioning behavior of RNA Pol II into phase-separated condensates that are associated with transcription initiation and elongation at genes driven by SEs [159]. In addition, N-terminal extension phosphorylation has been shown to promote NP1α oligomerization and conformational change to form heterochromatin-like droplets [160].

In contrast to transcriptional condensates, phase separation has also been implicated in the assembly of inactive chromatin compartments. For example, the repressive factor HP1α oligomers induce rapid chromatin compaction into nuclear puncta, suggesting that phase separation can be used to organize constitutive heterochromatin by recruiting known components of heterochromatin, such as nucleosomes, and DNA into HP1 droplets and selectively excluding TFIIB from the condensate [160]. Moreover, Plys et al. revealed that the canonical PRC1 phase-separated into dynamic puncta in cells, required the low-complexity disordered region of the CBX2 subunit, suggesting that PRC1 may facilitate the formation of condensates to compact the nucleosome and repress gene expression [161]. CBX2 also forms liquid-like condensates to concentrate DNA and nucleosomes through the highly charged positive residues within its IDR [162].

1,6-hexanediol treatment disrupts condensates by interrupting hydrophobic interactions [163]. This weakens the enhancer–promoter interactions and TAD insulation, but has little effect on CTCF- and cohesin-dependent loops, suggesting that LLPS suppression by 1,6-hexanediol did not affect the CTCF and cohesin binding across the genome. Decreased TAD insulation is likely caused by the effects of LLPS suppression and loop extrusion [164]. These results can also be postulated for YY1-mediated loops. 1,6-hexanediol affects many condensates and often requires high concentrations (0.1–1%, high mM), which may hinder use of this inhibitor as a therapeutic [165]. It is possible to design drugs through a physicochemical mechanism of action, a novel approach which suggests new concepts for traditionally undruggable cellular targets, such as intrinsically disordered proteins. For example, mitoxantrone, a planar heterocyclic small-molecule inhibitor, disrupts accumulation of amyotrophic lateral sclerosis-associated RNA-binding proteins (RBPs) in stress granules, possibly by interfering with the RNA-dependent assembly of a set of RBPs onto stress granules [166]. Another example is lipoamide, which has an inhibitory action that functions as a stress granule LLPS modulating inhibitor [167].

Interestingly, phase-separated condensates can concentrate small-molecule drugs and influence their pharmacodynamics by enhancing their target engagements [168]. Cisplatin, a platinum-based chemotherapy drug was found to be selectively concentrated in in vitro MED1 condensates, independent of its target DNA [168]. In HCT116 colon cancer cells, increased concentrations of cisplatin in MED1 condensates led to an increase of platinated DNA [168]. In cells, cisplatin treatment led to the preferential modification of SE DNA where MED1 was concentrated, and consequently resulted in dissolution of the condensates [168]. Given the ability of drugs to concentrate at specific condensates, it should be plausible to determine the chemical features of a small molecule that are responsible for selective condensate partitioning and to use this information to design small molecules with the ability to target specific cellular condensates. This effort could improve the therapeutic index of drugs and allow patients to be treated at lower doses with fewer side effects [169].

## 5. Conclusions and Future Directions

Cells rely on 3D genome organization for transcriptional control. Additionally, normal cells also utilize SEs to regulate transcription and form phase-separated condensates for their normal cellular functions. The critical question is to develop drugs that selectively target oncogenic SEs, specific chromatin interactions or cancer-associated condensates. Thus, an improved understanding of the mechanisms by which SEs and their transcriptional components promote oncogenic transcription and cancer-associated condensate formation could be useful in designing more selective and efficient drugs for cancer treatment.

Another critical question is how do we ensure that normal cells can tolerate 3D genome-modulating drugs, SE inhibitors and phase-separated condensate inhibitors? As 3D genome organization and liquid phase condensation are also critical processes in normal cells, we expect that drugs affect these processes might be toxic to normal cells as well as cancer cells. In this way, such drugs will be highly toxic to normal cells, causing a barrier to development in a clinical trial. Some drugs such as curaxins, can specifically kill cancer cells without severe toxicity for normal cells. However, the mechanism of such specificity needs to be addressed. Thus, in-depth research focusing on the differences in 3D genome organization in different types of cancers and normal cells need to be conducted.

The field of phase-separated condensates has received much interest in the past few years. Emerging evidence has shown that biomolecular condensates add new insights to the regulation of gene expression. Studies have shown that transcriptional activation in endogenous genomic loci could occur in the form of transactivation hub or phase separation [142,170,171]. Gene transcription typically involves *cis*-regulatory elements, such as an enhancer, to activate genes on the same chromosome [27]. In addition, *trans*-regulatory elements, such as extrachromosomal circular DNAs (ecDNAs) can promote intermolecular enhancer-gene interactions and drive oncogene amplification by recruiting RNA Pol II and BRD4 in ecDNA hubs [171,172,173]. Whether ecDNA can recruit a high number of TFs and transcriptional machinery to form transcriptional condensates through LLPS is still in question. In relation to that, the functional role of phase separation in gene transcription and gene repression is not entirely clear. More research is needed to understand what causes the difference between the nucleation of phase-separated condensates or transactivation hub at specific genomic loci to activate or silence genes. A combination of sequential fluorescence in situ hybridization and live-cell single-molecule imaging of protein interactions at physiological levels in live cells might be able to discern the difference between these two forms.

The formation of phase-separated condensate is a thermodynamic process, and, so far, most research has given much attention to the formation of condensates. The current view of how condensates regulate transcription activation is often simplistic. How such dynamically condensates achieve reversibility is another direction for future research. It is important to understand that the physicochemical properties determine condensate dynamics in cells in order to design better drugs that target specific condensates. To our knowledge, there is no study comparing the composition and dynamics of different phase-separated condensates in relation to transcription activation or repression.

Given the evidence that a gene can be activated by SEs in an additive [174], synergistic [175], redundant [176], or hierarchical manner [19], the mechanistic relationship among constituent enhancers is not clear in regulating transcriptional activation in condensates. As a typical enhancer can drive gene expression to a similar level to that of an enhancer cluster, it is not clear to what extent transcriptional condensates might differ between single strong enhancers and enhancer clusters [177]. It would also be interesting to understand how transcriptional condensates can mediate gene activation in higher-order genome architecture such as loops and TADs. Many experiments on condensates have been conducted using in vitro LLPS assays. These assays include expression and purification of proteins of interest, induction of LLPS of the purified proteins and studies of the biophysical properties of the liquids droplet formed by LLPS [178]. However, such in vitro assays cannot match the complexity of the cellular environment, this raises the question of how mechanisms differ between in vitro and endogenous genomic loci. Recent advances in microscopy at the nanoscale and live imaging studies of tagged loci, could be leveraged to address questions of the spatial organization of constituent enhancer elements with regards to each other and other regulatory elements at the single-allele level.

Concentrations and chemical modifications in RNA molecules have been shown to play a role in regulating condensate formation and dissociation [157,158]. However, the regulatory function of most long non-coding RNAs (lncRNAs) in relation to phase-separated condensation is unknown. A key feature of the most active ERα-bound enhancers is the recruitment of an ERα complex, referred to as the MegaTrans complex. It is composed of DNA-binding factors including RARγ/β, GATA3, AP2γ, AP1, FOXA1, and the DNA-dependent protein kinase. In response to 17β-estradiol, the MegaTrans complex is required for the activation of eRNA transcription and recruitment of p300 and MED1 [179]. These high local concentrations of eRNAs transcribed from MegaTrans enhancers, along with the components of MegaTrans complex, result in an eRNA-mediated ribonucleoprotein assembly exhibiting properties of phase-separated condensates in breast cancer [180], it will be interesting to determine whether altered expression of other lncRNAs that are commonly observed in cancer [181], mediate their oncogenic effects by regulating condensate formation. This has prompted further research to investigate the role of lncRNAs in relation to phase-separated condensates.

Condensate formation, dissolution, and function can be modified by the post-translational modifications of constituent components in the transcriptional condensates and are thought to change the selective condensate partitioning behaviour of molecules. Therefore, it is important to gain more insights into the epigenetic modifications of DNA, RNA and proteins that contribute to this selective partitioning behavior of these molecules. For example, third-generation sequencing platforms offered by Oxford Nanopore Technologies allow for direct identification of chemical modifications in DNA and RNA molecules [182,183,184], whereas tandem mass spectrometry can be used to map post-translational modifications for proteins of interest [185].

Estrogen enhances formation of MED1 condensates at the *MYC* oncogene in ER+ breast cancer cells and tamoxifen disrupts the formation of condensates [168]. ERα point somatic mutations (Y537S and D538G) reduced the affinity for tamoxifen in breast cancer cells [186]. Together, ERα D538G mutant protein and MED1 form condensates and these condensates were not disrupted upon addition of tamoxifen, suggesting that the concentration of tamoxifen in the condensate is inadequate to evict these ERα mutant proteins when the affinity is reduced [168]. However, it is not entirely clear how cancer-associated mutations play a role in condensate dysregulation with respect to the duration of estrogen and/or signal activation. It is tempting to make use of large databases of recurrent mutations to identify mutations that affect DNA, RNA and protein molecules whose functions are associated with condensates, including TFs, transcriptional co-activators, signaling proteins, cofactors, chromatin regulators, chromosome structuring proteins, and *cis*-elements (enhancers, promoters, and insulators). The pressing question is to determine the mechanisms by which these mutant molecules contribute to pathogenic condensate processes, and to identify the molecules that are involved therein that might become potential targets for cancer therapeutic intervention [169].

## Figures and Tables

**Figure 1 cancers-14-02866-f001:**
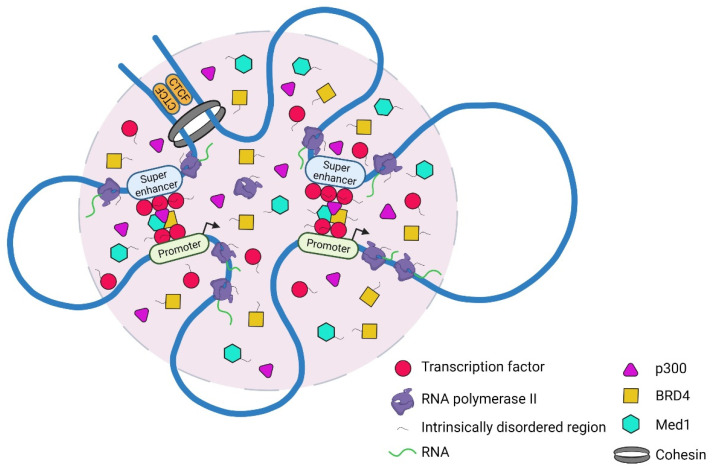
Accumulation of a high density of transcription factors, co-activators (BRD4, MED1, and p300), and RNA polymerase II at super-enhancers drives the formation of transcriptional condensates via phase separation. Transcriptional condensate formation is a cooperative process, involving a combination of weak multivalent protein–protein interactions and electrostatic protein–RNA interactions.

**Figure 2 cancers-14-02866-f002:**
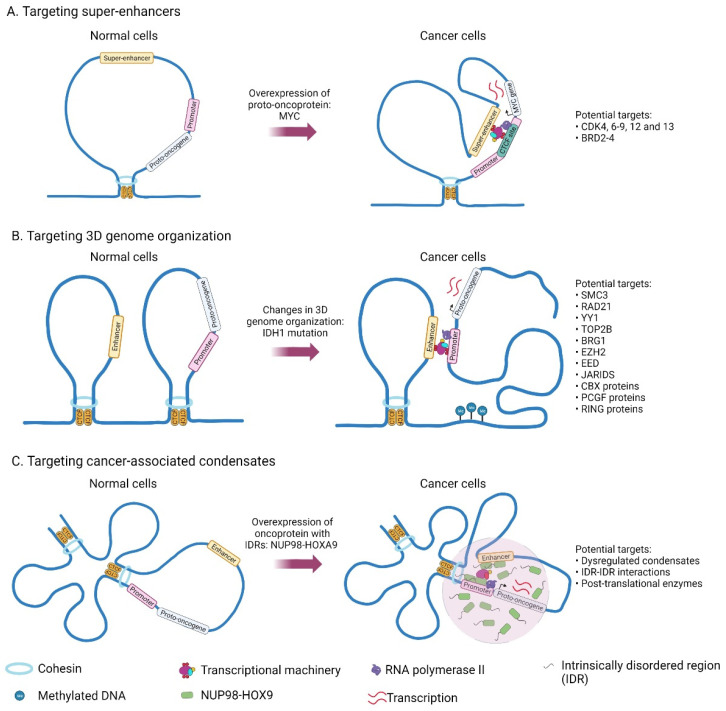
Genes and structures that are deregulated in cancer and which could potentially be targeted for cancer therapy.

**Table 1 cancers-14-02866-t001:** Targets and their potential inhibitors of disrupting SE components.

Target	Potential Small-Molecule Inhibitors	Reference
CDK7	THZ1, SY-1365, SY-5609, and THZ2	[50,61,62,63,64,65]
CDK4	Ribociclib (LEE011)	[6,66]
CDK6	Ribociclib (LEE011)	[6,66]
CDK12	THZ1, THZ531	[64,67]
CDK13	THZ1, THZ531	[64,67]
CDK8	Cortistatin A, SEL120-34A	[58,68]
CDK9	NVP-2	[64]
BRD2	I-BET762, OTX015, CPI0610, and BI-89499	[69,70,71]
BRD3	I-BET762, OTX015, CPI0610, and BI-89499	[69,70,71]
BRD4	JQ1, I-BET151, and I-BET762, OTX015, CPI0610, and BI-89499	[41,69,70,71,72]

**Table 2 cancers-14-02866-t002:** Targets and their potential inhibitors of disrupting 3D genome organization.

Target	Potential Small-Molecule Inhibitors	Effects in Cancer Hallmarks	Reference
SMC3	Glycyrrhizic acid, HDAC8-specific inhibitor PCI-34051	Delay cell cycle progression in MCF7 cells	[87,88]
RAD21	Glycyrrhizic acid	Inhibit Kaposi’s sarcoma-associated herpesvirus (KSHV)-infected cell growth	[87]
SMC1	No		
SA1 or SA2	No		
NIPBL	No		
WAPL	No		
YY1	siRNA YY1, nitric oxide donors, proteasome inhibitors, and inhibitors of activated survival pathways, such as inhibitors of nuclear factor-kappa beta	Inhibit cancer cell proliferation, viability and epithelial–mesenchymal transition	[91]
MAZ	No		
TOP2B	Doxorubicin, epirubicin, daunorubicin, idarubicin, mitoxantrone, etoposide, and mAMSA	Induce DNA damage and stop growth of cancer cells	[126]
ZNF143	No		
SNF2H	No		
TAF3	No		
BRG1	BRM/BRG1 ATP Inhibitor-1, PFI-3	Deprive of stemness and deregulated lineage specification for embryonic stem cells	[127,128]
EZH2	DZNep, EI1, EPZ005687, GSK343, GSK126, UNC1999, EPZ-6438, and Stabilized α-helix of EZH2 peptide (SAH-EZH2)	Inhibit cell growth, induce cell cycle arrest and apoptosis	[117]
EED	EED226, A-395, BR-001, EEDi-5285, EEDi-1056, MAK683, SAH-EZH2, Astemizole, Wedelolactone, and DC-PRC2in-01	The antitumor abilities of A-395 and GSK126 were validated in the Pfeiffer xenograft animal model	[119]
SUZ12	No		
RbAp46/48	No		
AEBP2	No		
PCLs	No		
JARIDS	Dihydroartemisinin	Inhibit cancer cell proliferation, migration, invasion, and tumor formation	[129]
CBX proteins	MS37452, UNC3866, MS37452, MS351, UNC4976, and SW2_152F	Block neuroendocrine differentiation and promote prostate cancer cell death	[130,131,132]
PCGF proteins	PTC209, QW24, PTC596, and PTC-028	Inhibit colorectal cancer cell proliferation, migration and self-renewal; and induce mitochondrial apoptosis in AML progenitor cells	[133,134,135,136]
HPH proteins	No		
RING proteins	RB-3, PRT4165	Induce differentiation in leukemia cell lines and primary AML samples; and inhibit DNA double-strand break repair and cause G2/M checkpoint failure	[137,138]

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
