# Peer review of "Super-Enhancers, Phase-Separated Condensates, and 3D Genome Organization in Cancer"

_cancers, 2022, doi:10.3390/cancers14122866_

Round 1

Reviewer 1 Report

The authors have undertaken a review of how super-enhancers and promoters interact via chromatin higher order structures.   The topic has been well written with a strong reference list.   My main concern is that Figure 1 is very late in the review and it would benefit the reader if either another figure was introduced earlier in the form a graphic showing the 3D genomic organisation, or perhaps moving the position of the current Figure 1.   A second figure could be added summarising what genes and structures are deregulated in cancer and who they could be targeted.   This review is for a Cancers journal not genomic structures journal so the readership should be considered. 

Reviewer 2 Report

Comments and Suggestions for Authors

The topic of the article is quite interesting and up to date. The authors summarized super-enhancers and phase-separated condensates, as well as their functions in the transcriptional activation of cancer-related genes and supported this with appropriate figure. The article is very interesting and deserves rapid publication. However, several concerns must be fully addressed before potential publication in Cancers.

  1. Line 62: Because enhancers and super-enhancers have comparable chromatin signatures, the authors must underline the degree of H3K27ac that distinguishes them.

  1. Line 77: At the DNA loop level, the authors should provide more information regarding DNA loop formation and provide the function and features of CTCF and cohesin in the loop extrusion model.

  1. In line 82: The authors introduce the concept of "insulated neighborhoods" without first providing a brief explanation of the genome's hierarchical architecture in 3D (also mentioned in the title). It would be useful to explain how super-enhancer promoter interactions are contained in insulated neighborhoods and maintained by CTCF and cohesin, AB compartments, chromosomal territories in general, etc. Pleas complete the information in this section.

  1. Lines 132: The title "Super-enhancers in Cancer" doesn't appear to fit the paragraph's information. Chromosome rearrangements, DNA mutations and indels, 3D chromatin structural alterations, and viral oncogenes are all mentioned as potential sources of super-enhancers by the authors. So a better title that relates would be "Mechanisms related to acquisition of super-enhancers in cancer", as example.

  1. Line 254: The authors explain the composition of proteins related to the architecture of the loops (CTCF, YY1 and cohesin components), I think that this information should be in the first pages of the review.

  1. The paragraph of line 238 "In particular, the disruption of TAD boundaries and dysregulated chromatin interactions can activate oncogene expression" would be desirable to include at the beginning of the topic of Super-enhancers in Cancer (line 132) to clarify that the disruption of TADs is due partly chromosomal rearrangements, mutation, SNPs, viral oncogenes etc. that are responsible for the aberrant promoter-SEs interactions that influence tumorigenic transcriptional programs.

  1. In table 2 different "Potential small-molecule inhibitors" are described, it is necessary to provide the related effects in cancer hallmarks an cellular processes such as cell cycle progression, suppressed proliferation and induced apoptosis, etc.

Author Response

Response to reviewers of manuscript Cancers-1738686

Reviewer 2:

The topic of the article is quite interesting and up to date. The authors summarized super-enhancers and phase-separated condensates, as well as their functions in the transcriptional activation of cancer-related genes and supported this with appropriate figure. The article is very interesting and deserves rapid publication. However, several concerns must be fully addressed before potential publication in Cancers.

  1. Line 62: Because enhancers and super-enhancers have comparable chromatin signatures, the authors must underline the degree of H3K27ac that distinguishes them.

Response to point 1: To distinguish super-enhancers from typical enhancers, we have added the following statements to page 3 line 75: “SEs have a higher order of magnitude of transcriptional factor density, size, and ability to activate transcription compared to the typical enhancer [4]. For example, although both SEs and typical enhancers are occupied by master TFs such as Oct4 and Nanog, SEs are more densely occupied by TFs, which are crucial for mESC identity such as Klf4. The histone modification levels at SEs exceeded the typical enhancers by at least an order of magnitude [4].”

  1. Line 77: At the DNA loop level, the authors should provide more information regarding DNA loop formation and provide the function and features of CTCF and cohesin in the loop extrusion model.

Response to point 2: For better clarification, we have added the following paragraph to page 3, line 89: “The human genome is organized into higher order structures, and such structures are important for transcriptional regulation [11]. Individual chromosomes occupy dis-tinct regions of the nucleus, known as chromosome territories, that are themselves spatially segregated in A and B compartments. The A compartment is associated with actively transcribed genes, whereas the B compartment is associated with epigenetically silent genes and gene-poor DNA. Genome-wide Hi-C analysis showed that loci located on the same chromosome interact more frequently than any two loci located on different chromosomes [12]. At the sub-megabase scale, chromatin is compart-mentalized into smaller structures known as topologically associating domains (TADs). TADs are self-interacting, loop-like domains that contain interacting cis-regulatory elements and target genes [13]. The chromatin fiber is organized into a collection of DNA loops which establish chromatin interactions with distant regions and regulate the activity of genes. This is explained by the loop extrusion model in which frequent transient loops are organised by structural maintenance of chromosomes (SMC) complexes that reel in chromatin, forming growing loops that stop at CCCTC-binding factor (CTCF) boundaries [14, 15]. TAD borders are demarcated by convergently oriented CTCF binding sites that obstruct loop extrusion and cohesin translocation. CTCF proteins act as loop anchors and insulate TADs from neighboring regions. Insulated neighborhoods are chromosomal loops, which bound by CTCF homodimers, occupy by the cohesin complex, and contain at least one gene [16, 17]. Most of the enhancer-promoter interactions are contained within insulated neighborhoods [18].”

  1. In line 82: The authors introduce the concept of "insulated neighborhoods" without first providing a brief explanation of the genome's hierarchical architecture in 3D (also mentioned in the title). It would be useful to explain how super-enhancer promoter interactions are contained in insulated neighborhoods and maintained by CTCF and cohesin, AB compartments, chromosomal territories in general, etc. Please complete the information in this section.

Response to point 3: We have addressed the reviewer’s comments on point 2 and point 3 by combining the statements to introduce the hierarchical structure of 3D genome organization in the order of chromosome territories, AB compartments, TADs and chromatin loops. In this paragraph, we also introduce the loop extrusion model and insulated neighborhoods in page 3 line 89.

  1. Lines 132: The title "Super-enhancers in Cancer" doesn't appear to fit the paragraph's information. Chromosome rearrangements, DNA mutations and indels, 3D chromatin structural alterations, and viral oncogenes are all mentioned as potential sources of super-enhancers by the authors. So a better title that relates would be "Mechanisms related to acquisition of super-enhancers in cancer", as example.

Response to point 4: We have changed the title for section 2.2 to “Mechanisms related to the acquisition of Super-enhancers in Cancer” in page 4 line 164.

  1. Line 254: The authors explain the composition of proteins related to the architecture of the loops (CTCF, YY1 and cohesin components), I think that this information should be in the first pages of the review.

Response to point 5: We have added the following statement to the abstract of this manuscript on page 1 line 23: “The 3D genome is highly maintained by several architectural proteins, such as CTCF, Yin Yang 1 and cohesin complex.”

  1. The paragraph of line 238 "In particular, the disruption of TAD boundaries and dysregulated chromatin interactions can activate oncogene expression" would be desirable to include at the beginning of the topic of Super-enhancers in Cancer (line 132) to clarify that the disruption of TADs is due partly chromosomal rearrangements, mutation, SNPs, viral oncogenes etc. that are responsible for the aberrant promoter-SEs interactions that influence tumorigenic transcriptional programs.

Response to point 6: We have added the following statement to page 4, line 167: “In particular, the disruption of TAD boundaries and dysregulated chromatin interactions can activate oncogene expression”.

  1. In table 2 different "Potential small-molecule inhibitors" are described, it is necessary to provide the related effects in cancer hallmarks an cellular processes such as cell cycle progression, suppressed proliferation and induced apoptosis, etc.

Response to point 7: We have added an additional column labeled “Effects in cancer hallmarks” into Table 2 to clarify the cellular processes that are targeted by the inhibitors.